# Engagement with tNOX (ENOX2) to Inhibit SIRT1 and Activate p53-Dependent and -Independent Apoptotic Pathways by Novel 4,11-Diaminoanthra[2,3-*b*]furan-5,10-diones in Hepatocellular Carcinoma Cells

**DOI:** 10.3390/cancers11030420

**Published:** 2019-03-24

**Authors:** Chia-Yang Lin, Atikul Islam, Claire J. Su, Alexander S. Tikhomirov, Andrey E. Shchekotikhin, Show-Mei Chuang, Pin Ju Chueh, Yao Li Chen

**Affiliations:** 1Institute of Biomedical Sciences, National Chung Hsing University, Taichung 40227, Taiwan; lcy4534@gmail.com (C.-Y.L.); islammiu555@gmail.com (A.I.); suc1@mca.org.tw (C.J.S.); smchuang@dragon.nchu.edu.tw (S.-M.C.); 2Morrison Academy in Taichung, 216 Si Ping Road, Taichung 40679, Taiwan; 3Gause Institute of New Antibiotics, 11B. Pirogovskaya Street, 119021 Moscow, Russia; tikhomirov.chem@gmail.com (A.S.T.); shchekotikhin@mail.ru (A.E.S.); 4Department of Organic Chemistry, Mendeleyev University of Chemical Technology, 9 Miusskaya Square, 125047 Moscow, Russia; 5Graduate Institute of Basic Medicine, China Medical University, Taichung 40402, Taiwan; 6Department of Medical Research, China Medical University Hospital, Taichung 40402, Taiwan; 7Department of General Surgery, Changhua Christian Hospital, Changhua 50008, Taiwan; 8School of Medicine, Kaohsiung Medical University, Kaohsiung 80708, Taiwan

**Keywords:** apoptosis, anthraquinone derivatives, hepatocellular carcinoma, p53, tumor-associated NADH oxidase (tNOX or ENOX2), sirtuin 1 (SIRT1), c-Myc, protein acetylation

## Abstract

Hepatocellular carcinoma (HCC) is the most frequent primary malignancy of the liver and is among the top three causes of cancer-associated death worldwide. However, the clinical use of chemotherapy for HCC has been limited by various challenges, emphasizing the urgent need for novel agents with improved anticancer properties. We recently synthesized and characterized a series of 4,11-diaminoanthra[2,3-*b*]furan-5,10-dione derivatives that exhibit potent apoptotic activity against an array of cancer cell lines, including variants with multidrug resistance. Their effect on liver cancer cells, however, was unknown. Here, we investigated three selected 4,11-diaminoanthra[2,3-*b*]furan-5,10-dione derivatives (compounds **1**–**3**) for their cytotoxicity and the underlying molecular mechanisms in wild-type or p53-deficient HCC cells. Cytotoxicity was determined by WST-1 assays and cell impedance measurements and apoptosis was analyzed by flow cytometry. The interaction between compounds and tumor-associated NADH oxidase (tNOX, ENOX2) was studied by cellular thermal shift assay (CETSA). We found that compound **1** and **2** induced significant cytotoxicity in both HepG2 and Hep3B lines. CETSA revealed that compounds **1** and **2** directly engaged with tNOX, leading to a decrease in the cellular NAD^+^/NADH ratio. This decreased the NAD^+^-dependent activity of Sirtuin 1 (SIRT1) deacetylase. In p53-wild-type HepG2 cells, p53 acetylation/activation was enhanced, possibly due to the reduction in SIRT1 activity, and apoptosis was observed. In p53-deficient Hep3B cells, the reduction in SIRT1 activity increased the acetylation of c-Myc, thereby reactivating the TRAIL pathway and, ultimately leading to apoptosis. These compounds thus trigger apoptosis in both cell types, but via different pathways. Taken together, our data show that derivatives **1** and **2** of 4,11-diaminoanthra[2,3-*b*]furan-5,10-diones engage with tNOX and inhibit its oxidase activity. This results in cytotoxicity via apoptosis through tNOX-SIRT1 axis to enhance the acetylation of p53 or c-Myc in HCC cells, depending on their p53 status.

## 1. Introduction

Hepatocellular carcinoma (HCC) is the most frequent primary malignancy of the liver and is among the top three most common causes of cancer-associated death worldwide [1,2]. The treatment options for HCC patients include surgical resection, liver transplantation, percutaneous radiofrequency ablation, and either with or without associated chemotherapy [3]. Chemotherapeutic agents bearing an anthracene-9,10-dione moiety, such as doxorubicin, mitoxantrone, valrubicin, etc., are currently used as first-line treatments for HCC patients; the above-named moiety of these agents can interact with DNA duplexes to inhibit topoisomerase 2 (Topo2) and provoke apoptosis [4,5]. However, the clinical use of doxorubicin has been limited by its significant oxidative stress-associated toxicity, low efficacy, and tendency to trigger acquired resistance [6,7]. Thus, it is urgent that we search for and identify novel pharmacological drugs with improved anticancer properties.

We recently reported the synthesis of a series of novel anti-cancer compounds based on furan-fused anthraquinone derivatives that have different side chains, and showed that these derivatives have high antiproliferative activities against various cancer cell lines, including some with resistance to doxorubicin [8,9,10,11,12]. Biological studies revealed that these derivatives exert cytotoxicity via multiple mechanisms, such as by down-regulating a tumor-associated NADH oxidase (tNOX, ENOX2) and Sirtuin 1 (SIRT1) [8]. The human tNOX gene, located on Xq25-26, encodes a protein of 610 amino acids and is universally expressed in an array of cancer/transformed cells derived from solid tumors [13,14,15,16]. Overexpression of tNOX in non-cancer cells and its depletion in cancer cells confirmed that tNOX is essential for and positively associated with cell proliferation and migration [17,18,19,20,21]. In cancer therapy, it is thought that the suppression of tNOX by various agents (e.g., capsaicin, epigallocatechin gallate, tamoxifen, oxaliplatin) induces apoptosis and attenuates cancer cell growth [19,22,23,24,25]. tNOX converts reduced NADH or hydroquinones to oxidized NAD^+^ [14,15,26], and recent studies have demonstrated that inhibition of tNOX activity reduces the intracellular NAD^+^ level, which in turn impacts NAD^+^-dependent SIRT1 deacetylase activity and apoptosis [8,25,27].

Although a close correlation between tNOX inhibition and apoptosis has been established in many cancer cell lines, it has not previously been studied in HCC cells. Here, we explored a new therapeutic strategy for HCC by testing the effectiveness of three selected 4,11-diaminoanthra[2,3-*b*]furan-5,10-diones (compounds **1**–**3**) on p53 wild-type HepG2 and p53-deficient Hep3B cancer cell lines. Given that p53 triggers both cell-autonomous and non-cell-autonomous mechanisms to suppress tumor development in HCC [28], it is not surprising p53 is considered to be a key target for treating HCC. Mutation and misregulation of p53 also leads to an acquired resistance that considerably limits the effectiveness of therapeutic drugs for cancer management. To address these issues, researchers are seeking to identify additional factors in the hopes of designing new therapeutic strategies. Here, using a cellular thermal shift assay (CETSA), we demonstrate that compounds **1** and **2** engage with tNOX and inhibit SIRT1 to induce apoptosis against both p53-wild-type and p53-deficient HCC cell lines.

## 2. Results

### 2.1. 4,11-Diaminoanthra[2,3-b]furan-5,10-diones **1** and **2** Suppress the Growth of HCC Cells

We recently reported the synthesis of new anticancer anthra[2,3-*b*]furan-5,10-dione derivatives with different 4,11 side chains, showed that they potently inhibited the proliferation of various cancer cell lines, including some multidrug resistance sublines with deletion of the p53 gene, and further demonstrated that the cytotoxicity of these agents in cancer cells may be associated with the inhibition of tNOX [8]. Given that HCC is one of the top three most common causes of cancer-associated death worldwide, we herein investigated the antiproliferative potency of our new compounds in liver cancer cells and assessed the mechanisms underlying the ability of these compounds to induce apoptosis. We focused on 4,11-diaminoanthra[2,3-*b*]furan-5,10-dione derivatives **1**–**3** (Figure 1), which bear ((2-dimethylamino)ethyl)amino, 2-((2-hydroxyethyl)amino)ethylamino, and 2-(guanidino)ethylamino groups, respectively, at their 4,11-positions.

These derivatives were synthesized in accordance with our previously described scheme [8]. Cell viability assays revealed that compounds **1** and **2** significantly inhibited the proliferation of HepG2 cells, whereas the derivative bearing terminal guanidine groups on its side chains (compound **3**) exhibited no cytotoxicity in this cell line (Figure 2A). Similar results were obtained using cell impedance, which demonstrated that HepG2 cell growth was markedly inhibited by compounds **1** and **2** at 2 μM over a 70 h period (Figure 2B). Interestingly, compounds **1** and **2** also significantly attenuated cell viability and cell growth in p53-deficient HCC Hep3B cells (Figure 2C,D)

These cytotoxic derivatives had little impact on generation of reactive oxygen species (ROS), especially in Hep3B cells, as determined using the fluorescent dye, H_2_DCFDA (Figure 2E). The lack of ROS generation in treated HCC cells suggests that the compounds could potentially induce less oxidative stress-mediated cardiomyopathy, which is an adverse effect often found in patients treated with doxorubicin and similar agents [29,30].

### 2.2. 4,11-Diaminoanthra[2,3-b]furan-5,10-diones **1** and **2** Bind to tNOX and Inhibit tNOX-Catalyzed NAD^+^ Generation

In our previous study, we demonstrated that the newly synthesized 4,11-diaminoanthra[2,3-*b*]furan-5,10-diones induced apoptosis through tNOX down-regulation; however, the detailed molecular mechanism was not explored [8]. Here, we found that tNOX expression was slightly reduced by compounds **1** and **2**, but not by compound **3**, in HepG2 and Hep3B cells (Figure 3A). Given that ligand binding normally enhances protein stability [31,32,33], we employed a cellular thermal shift assay (CETSA) to investigate the interaction of the compounds with tNOX protein. In our CETSA, tNOX proteins incubated with or without compounds were collected, incubated at different temperatures, and recognized by antisera to tNOX on western blots, which enabled us to plot thermal melting curves and derive the melting temperature (*T*_m_; the temperature at which 50% of proteins are unfolded and rapidly precipitated by heat). We found that anthra[2,3-*b*]furan-5,10-dione **2** elicited an obviously different shift in the thermal melting curves of treated lysates versus untreated lysates from HCC cells. In HepG2 cells, the *T*_m_ value increased from 49.1 °C (control) to 54.5 °C (treated with compound **2**), suggesting that compound **2** engaged with tNOX protein and enhanced its thermal stability (Figure 3B). In p53-deficient Hep3B cells, there was also a marked shift in the *T*_m_ values between lysates treated with compound **2** and untreated control lysates (Figure 3C) and a greater than 5 degree difference in the *T*_m_ of cells treated with and without compound **1** (Figure 3D). In contrast, compound **3** had little effect (less than 5 degrees) on the thermal melting curves of treated versus untreated cell lysates, suggesting that this compound is incapable of engaging with the tNOX protein in Hep3B cells (Figure 3E). As tNOX catalyzes the conversion of reduced NADH to oxidized NAD^+^ [14,15], it is relevant that the binding of compound **2** to the tNOX protein resulted in a marked and dose-dependent attenuation of the intracellular NAD^+^/NADH ratio in Hep3B cells (Figure 3F). These findings suggest that there is a direct link between the cytotoxicity of anthra[2,3-*b*]furan-5,10-dione compound **2** and its ability to inhibit tNOX oxidase activity.

### 2.3. 4,11-Diaminoanthra[2,3-b]furan-5,10-diones **1** and **2** Inhibit SIRT1 Deacetylase Activity and Enhance p53/c-Myc Acetylation, Leading to Apoptosis, Not Autophagy

As the above results indicated that compound **2** slightly attenuated tNOX expression and reduced the intracellular NAD^+^/NADH ratio by engaging with tNOX in HCC cells, we next sought to understand the cellular outcomes of these changes. As NAD^+^ is a cofactor for SIRT1 deacetylase activity, we hypothesized that the compound-induced variation in the intracellular NAD^+^/NADH ratio might impact SIRT1 and subsequent cellular events. Our investigations revealed that SIRT1 expression was somewhat diminished by compounds **1** and **2**, and that p53 acetylation was markedly enhanced in HepG2 cells exposed to compound **2**, possibly due to the reduction in SIRT1 deacetylase activity (Figure 4A). The enhanced p53 acetylation was accompanied by increases in p53 phosphorylation, anti-survival NOXA, PUMA, p21, Bak expression, and PARP cleavage, as well as down-regulation of anti-apoptotic Bcl2, all of which are suggestive of apoptosis (Figure 4A). Indeed, a prominent induction of apoptosis by compound **2** was confirmed by Annexin V-FITC/PI staining followed by flow cytometry (Figure 4B).

In p53-deficent Hep3B cells, our flow cytometric data confirmed that compounds **1** and **2**, but not compound **3**, triggered marked apoptosis independent of p53 in Hep3B cells (Figure 5A). We also validated that treatment with compounds **1** and **2** led to slight reduction in SIRT1 expression; moreover, c-Myc acetylation was noticeably enhanced by aminoalkylamino-bearing derivatives **1** and **2**, but not guanidine-bearing compound **3**, possibly through the ability of the former to suppress SIRT1 deacetylase activity (Figure 5B). The alteration in c-Myc acetylation was accompanied by up-regulation of TRAIL, death receptor 5 (DR5), and pro-apoptotic Bak, down-regulation of pro-survival c-Flip and Bcl-2, and increased caspase 3-directed PARP-cleavage, all of which are consistent with the induction of apoptosis. Interestingly, apoptosis induced by compounds **1** and **2** was partially blocked by co-treatment with TRAIL-neutralizing antibody, suggesting that these two derivatives triggered TRAIL-death receptor apoptosis signaling pathway (Figure 5C). Conversely, autophagy appears to play an insignificant role in the cytotoxicity induced by these derivatives, even at high concentrations of 10 μM (Figure 5D). Lack of marked induction of cleaved LC3-II also supported that compounds **1** and **2** are not autophagic (Figure 5E). These various lines of evidences support the notion that the two tested novel anthra[2,3-*b*]furan-5,10-diones inhibit SIRT1 deacetylase activity through p53-dependent and -independent pathways to induce apoptosis.

### 2.4. tNOX Expression Is Essential for Hep3B Cell Survival and Correlates with Tumor Progression and Poor Prognosis in an Online Database

The involvement of tNOX expression in liver cancer cell survival was next evaluated utilizing RNA interference (RNAi) technology in p53-deficient Hep3B cells. We found that transfection with tNOX-specific siRNA effectively reduced tNOX expression compared to that seen in control cells, and that tNOX depletion significantly enhanced spontaneous apoptosis in Hep3B cells compared to untreated control cells (Figure 6).

We further explored the relationship of tNOX (ENOX2) expression in liver hepatocellular carcinoma survival outcomes by data mining in Kaplan-Meier plotter (pan-cancer RNA-seq dataset; www.kmplot.com). Among 249 male patients with liver cancer, high tNOX expression appeared to be associated with a poor prognosis for overall survival [hazard ratio (HR): 1.76, log-rank *p* = 0.015]. The median overall survival in the low tNOX expression cohort was 82.87 months, compared to 38.3 months in the high tNOX expression cohort (Figure 7A).

An insignificant correlation was found between high tNOX expression and decreased overall survival in 121 female patients (HR: 1.87, log-rank *p* = 0.069), but the median survival was much lower in the high tNOX expression cohort (37.83 months) compared to the low tNOX expression cohort (70.53 months) (Figure 7B). Notably, low tNOX expression was significantly correlated with better survival outcomes if the follow-up threshold was adjusted to 60 months in the same female population (HR: 2.32, log-rank *p* = 0.036). The upper quartile survival was 40.33 months in the low tNOX expression cohort, which was significantly longer than the 18.5 months seen in the low tNOX expression cohort (Figure 7C). Beyond HCC, a striking negative correlation between tNOX expression and prognosis was found in female patients with breast cancer (HR: 1.72, log-rank *p* = 0.0096) (Figure 7D).

Together, our various lines of evidence suggest that tNOX expression is positively correlated with worse survival outcomes, and that this high expression can be suppressed by anti-neoplastic 4,11-diaminoanthra[2,3-*b*]furan-5,10-diones or RNA interference to induce apoptosis. The novel derivatives tested herein exert their apoptotic activity by engaging with tNOX to reduce NADH oxidation, which inhibits SIRT1 deacetylase activity. In p53 wild-type HCC cells, this activates p53; in p53-deficent Hep3B cells, it increases c-Myc acetylation to up-regulate TRAIL expression, which also leads to apoptosis (Figure 8).

## 3. Discussion

We report that two novel 2-unsubstituted 4,11-diaminoanthra[2,3-*b*]furan-5,10-dione derivatives, herein called compounds **1** and **2**, induce potent cytotoxicity in human HCC cells by triggering apoptosis. We show that the apoptotic potency of these derivatives results from their engagement with tNOX, a tumor-associated NADH oxidase, which in turn reduces the intracellular NAD^+^/NADH ratio. This suppresses NAD^+^-dependent SIRT1 deacetylase activity, leading to enhanced p53 acetylation/activation and apoptosis in HepG2 cells. Alternatively, in p53-deficient Hep3B cells, we observe increased c-Myc acetylation; this up-regulates TRAIL, which is essential for apoptosis induction (Figure 8). Our results collectively suggest that anthrafurandione compounds **1** and **2**, whose side chains resemble those of the mitoxantrone, ((2-dimethylamino)ethyl)amino and 2-((2-hydroxyethyl)amino)ethylamino, respectively, show antiproliferative potency against human HCC cells. The clinical use of anthracycline chemotherapeutics, including doxorubicin, in cancer treatment has been complicated by cardiotoxicity [34,35,36]. Although there is still debate on this matter, many lines of evidence indicate that anthracycline-mediated cardiotoxicity is associated with mitochondrial abnormalities and oxidative stress [37,38]. In this context, it is notable that compounds **1** and **2** failed to induce high-level ROS generation in Hep3B cells; thus, their clinical use may result in fewer cardiomyopathic adverse effects. In contrast to compounds **1** and **2**, compound **3**, which bore guanidinated terminal amino groups, had far less cytotoxic potential, probably due to its overly high positive charge and subsequent reduced binding with intracellular targets. Indeed, we observed a correlation between low tNOX engagement and the absence of cytotoxicity for compound **3**. It is also possible that the introduction of guanidine resides in the side chains of anthrafurandiones and related anthraquinones may decrease their uptake and cellular distribution [8,10,11].

An array of well-known anti-cancer agents, including capsaicin, tea catechin, doxorubicin, and phenoxodiol, have been shown to exert their anti-neoplastic activities by inhibiting or down-regulating tNOX to induce apoptosis [19,22,23,24,39,40]. However, no previous work has shown that tNOX engages with such drugs to achieve its therapeutic effects. Here, we used CETSA to directly assess the changes in tNOX upon exposure of anthra[2,3-*b*]furan-5,10-diones **1** and **2** but not **3**. After heating, it appeared that the treatment of cells with compounds **1** or **2** prevented tNOX from becoming denatured and precipitated; instead, the ligand-bound complexes remained in solution to be visualized by western blot analysis. This engagement inhibited the ability of tNOX to oxidize the reduced form of NADH, leading to decreases in NAD^+^ generation and SIRT1 deacetylase activity, which ultimately resulted in apoptosis and inhibition of cancer cell growth. Similar findings by another study [41] clearly suggest that tNOX-oxaliplatin binding interferes with its oxidase activity and plays a critical role in regulating apoptosis. Using in vitro cell models, we confirmed that tNOX depletion restored non-cancer phenotypes, including abrogated anchorage-independent growth, increased sensitivity to stress-mediated apoptosis, attenuated migration, and prolonged cell cycle progression, further supporting the importance of tNOX in cancer management [17,19,21,25,42].

Another significant contribution of this work is our evidence showing that the tested derivatives activate differential cellular responses to induce apoptosis in p53-dependent and -independent fashions. In p53-wild-type HepG2 cells, we found that binding between tNOX and compound **2** reduced the intracellular NAD^+^ concentration, resulting in SIRT1 inhibition and enhancement of p53 acetylation, which in turn up-regulated its downstream target, pro-apoptotic Bak. Consistent with our results, the increased p53 acetylation that follows SIRT1 inhibition was shown to be associated with an induction of PUMA, which up-regulates Bak and triggers apoptosis [43]. Inspired by a recent study revealing that acetylated c-Myc loses its ability to impair TRAIL expression [44], we also investigated c-Myc. Indeed, in p53-deficient Hep3B cells, we found that compounds **1** and **2** both enhanced c-Myc acetylation with concurrent TRAIL up-regulation and apoptosis. Our finding of increased apoptosis in Hep3B cells treated with compounds **1** and **2** is consistent with a previous report showing that the inhibition of SIRT1 is linked to DR5 up-regulation and c-Flip suppression, which sensitizes human leukemic K562 cells to apoptosis [45]. Interestingly, c-Myc has been shown to induce SIRT1 at the transcriptional or posttranscriptional levels, and SIRT1 was found to interact with and deacetylate c-Myc [46,47]. Although there is some debate regarding how SIRT1-mediated deacetylation affects the stability of c-Myc, it is clear that a c-Myc-SIRT1 feedback loop significantly impacts cell growth and transformation [46,47,48,49]. Our present results agree with the previous findings by showing that the SIRT1-inhibition-mediated enhancement of c-Myc acetylation is accompanied by up-regulation of TRAIL and DR5, supporting the idea that SIRT1 plays a tumor-promoting role in the context of c-Myc activation [50,51].

It has been well documented that tNOX positively regulates cell proliferation and migration [17,18,19,20,21], but much less is known regarding the expression and the related clinical outcomes. Previous work showed that anti-cancer drug treatments suppressed the tNOX activity observed in pooled sera of cancer patients but not non-cancer-bearing volunteers [52,53,54]. Using data mining in Kaplan-Meier plotter (www.kmplot.com), we herein observed that the overall survival in 370 patients with liver cancer tended to be correlated with tNOX expression, although not to a significant degree. The median overall survival in the low tNOX expression cohort was 61.73 months, compared to 46.2 months in the high tNOX expression cohort (HR: 1.37, log-rank *p* = 0.11). When we separated the patients by gender, we found that tNOX expression was significantly correlated with poor prognosis in male patients, but not in female patients (Figure 7A,B), suggesting that gender is a key factor that affects survival outcomes. This finding is not surprising, given that the tNOX gene resides on the X chromosome [13] and previous work showed that gender influences the survival of colorectal cancer patients [55,56]. Thus, the gender-related differences in clinical outcomes may reflect a dosage effect on genes expressed from sex chromosomes [57]. Using the online EMBL-EBI database (http://www.ebi.ac.uk/gxa), we found that tNOX (ENOX2) expression was 2 Log2-fold enhanced in specimens from primary colorectal tumor patients with and without distant metastasis, as assessed in a microarray analysis [58]. Interestingly, a previous study found that the use of hepatocyte growth factor to induce the epithelial-mesenchymal transition (EMT) was associated with significant induction of tNOX expression in small cell lung cancer cells [59]. This was consistent with our previous proposal that up-regulation of tNOX enhances cell migration and EMT [24]. tNOX expression has also been positively associated with lymphangiogenesis, and high-level tNOX-expression was found to be associated with very metastatic phenotypes in a population of melanoma patients [60]. These converging lines of evidence strongly suggest that tNOX must be suppressed in order for a patient to have a better prognosis.

In sum, we herein used CETSA to show that tNOX directly engages with two novel anthraquinone derivatives to mediate different pathways that can lead to apoptosis even in a p53-inactivated system. Our experimental findings and the results of our clinical data mining collectively indicate that tNOX could be a useful therapeutic drug target. These novel anthraquinone-derived compounds inhibit tNOX-NAD^+^-SIRT1 axis to induce apoptosis, highlighting the value of tNOX as a possible therapeutic target and implying future clinical use of these compounds in cancer treatment without considering p53 expression.

## 4. Materials and Methods

### 4.1. Materials

Fetal bovine serum (FBS) and penicillin/streptomycin were obtained from Gibco/BRL Life Technologies (Grand Island, NY, USA). The anti-Bak, anti-Bcl-2, anti-PARP, anti-p53, anti-phospho-p53, anti-acetyl-p53, anti-Noxa, anti-PUMA, anti-TRAIL, anti-DR5, anti-C-Flip and anti-SIRT1 antibodies were purchased from Cell Signaling Technology, Inc. (Beverly, MA, USA). The anti-p21 and anti-Myc antibody were purchased from Santa Cruz Biotechnology, Inc. (Santa Cruz, CA, USA). The anti-acetyl-Myc and β-actin antibody were from EMD Millipore, Inc. (Burlington, MA, USA). The antisera to tNOX used in our western blot analyses were generated as described previously [18]. The anti-mouse and anti-rabbit IgG antibodies and other chemicals were purchased from Sigma Chemical Company (St. Louis, MO, USA) unless otherwise specified.

### 4.2. Chemistry

4,11-Diaminoanthra[2,3-*b*]furan-5,10-dione derivatives **1**–**3** were prepared in accordance with the previously described procedure [8]. The purities of the tested samples of **1**–**3** were >95%, as determined by HPLC analysis.

### 4.3. Cell Culture and Transfection

HepG2 (p53 wild-type) (The Bioresource Collection and Research Center, BCRC, Hsinchu, Taiwan) and Hep3B (p53-deficent, derived from human HCC) (ATCC, Manassas, VA, USA) were grown in MEM supplemented with 10% fetal bovine serum, 100 units/mL penicillin and 50 µg/mL streptomycin. The cells were grown at 37 °C in a humidified atmosphere of 5% CO_2_ in air, with replacement of the medium every 2–3 days. The experimental groups were treated with different concentrations of the test compounds dissolved in DMSO, and the controls were treated with the same volume of DMSO.

ON-TARGETplus tNOX (ENOX2) siRNA and negative control siRNA were purchased from Thermo Scientific, Inc. (Grand Island, NY, USA). Briefly, cells were seeded in 10-cm dishes, allowed to attach overnight, and then transfected with tNOX siRNA or control siRNA using the Lipofectamine RNAiMAX Reagent (Gibco/BRL Life Technologies) according to the manufacturer’s instructions.

### 4.4. Continuous Monitoring of Cell Impedance

For continuous monitoring of changes in cell growth, cells (7.5 × 10^3^ cells/well) were seeded onto E-plates and incubated for 30 min at room temperature. The E-plates were placed onto a Real-Time Cell Analysis (RTCA) station (ACEA Biosciences, Inc. San Diego, CA, USA) and the cells were grown overnight before being exposed to test compounds or ddH_2_O. Cell impedance was defined by the cell index (CI) = (Z_i_ − Z_0_) [Ohm]/15[Ohm], where Z_0_ is the background resistance and Z_i_ is the resistance at an individual time point. CI readouts were initiated after the cell seeding and measurements were taken every hour for a total of 72 h. The results are reported as normalized CI values, which were determined as the CI at a certain time point divided by that at the beginning of the exposure.

### 4.5. Cell Viability Assay

Cells (5 × 10^3^) were seeded in 96-well culture plates in medium containing 10% serum, incubated overnight at 37 °C, and treated with test compounds. At the end of the treatment period, cell viability was determined using the WST-1 reagent (Roche) as described by the manufacturer. All experiments were performed at least in triplicate on three separate occasions. Data are presented as means ± SDs.

### 4.6. Apoptosis Determination

Apoptosis was measured using an Annexin V-FITC apoptosis detection kit (BD Pharmingen, San Jose, CA, USA). For the TRAIL-neutralization experiment, cells were pre-treated with IgG or anti-TRAIL antibody (Cell Signaling Technology) to a final concentration of 1 μL/mL for one h before exposed to different derivatives for 24 h. Cells cultured in 6-cm dishes were trypsinized, collected by centrifugation, washed, resuspended in 1× binding buffer, and stained with Annexin V-FITC, as recommended by the manufacturer. Cells were also stained with propidium iodide (PI) for detection of necrosis or late apoptosis. The distributions of viable (FITC/PI double-negative), early apoptotic (FITC-positive), late apoptotic (FITC/PI double-positive), and necrotic (PI-positive/FITC-negative) cells were analyzed using a FC500 flow cytometer (Beckman Coulter, Inc., Indianapolis, IN, USA). The results are expressed as a percentage of total cells.

### 4.7. Measurement of Intracellular NAD^+^/NADH Ratio

The oxidized and reduced forms of intracellular NAD were determined using an NADH/NAD Quantification Kit (BioVision Inc., Milpitas, CA, USA), as described by the manufacturer. Briefly, 2 × 10^5^ cells were washed with cold PBS, pelleted, and extracted by two freeze/thaw cycles with 400 µL of NADH/NAD^+^ extraction buffer. The samples were vortexed and centrifuged at 14,000 rpm for 5 min. The extracted NADH/NAD^+^ supernatant (200 µL) was transferred to a microcentrifuge tube, heated to 60 °C for 30 min (to decompose NAD^+^ but not NADH), and placed on ice. The samples were then centrifuged and transferred to a multiwell-plate. Standards and a NAD^+^ cycling mix were prepared according to the manufacturer’s protocol. The reaction mix (100 µL) was distributed to each well containing NADH standards and samples, and the plates were incubated at room temperature for 5 min to convert NAD^+^ to NADH. The provided NADH developer solution was dispensed to each well, and plates were incubated at room temperature for 15 or 30 min. The reaction was stopped with 10 µL of stop solution per well, and absorbance was measured at 450 nm.

### 4.8. Cellular Thermal Shift Assay (CETSA)

Engagement between each compound and tNOX in cells was analyzed by CETSA. Samples were prepared from control and drug-exposed cells. For each set, 2 × 10^7^ cells were seeded in a 10-cm cultured dish. After 24 h of culture, the cells were pretreated with 10 μM MG132 for 1 h, washed with PBS, treated with trypsin, and collected. Samples were centrifuged at 12,000 rpm for 2 min at room temperature, the pellets were gently resuspended with 1 mL of PBS, and the samples were centrifuged at 7500 rpm for 3 min at room temperature. The pellets were resuspended with 1 mL of PBS containing 20 mM Tris-HCl pH 7.4, 100 mM NaCl, 5 mM EDTA, 2 mM phenylmethylsulfonyl fluoride (PMSF), 10 ng/mL leupeptin, and 10 μg/mL aprotinin. The samples were transferred to Eppendorf tubes and subjected to three freeze-thaw cycles; for each cycle, they were exposed to liquid nitrogen for 3 min, placed in a heating block at 25 °C for 3 min, and vortexed briefly. The samples were then centrifuged at 12,000 rpm for 30 min at 4 °C, and the supernatants were transferred to new Eppendorf tubes. For the experimental sample set, each test compound was added to a final concentration of 100 μM; for the control sample set, the same volume of vehicle solvent was added. The samples were heated at 37 °C for 1 h and dispensed to 100 μL aliquots. Pairs consisting of one control aliquot and one experimental aliquot were heated at 40 °C, 43 °C, 46 °C, 49 °C, 52 °C, 55 °C, 58 °C, 61 °C, or 67 °C for 3 min. Finally, the samples were placed on ice and subjected to western blot analysis using antisera to tNOX [18,61].

### 4.9. Measurement of Reactive Oxygen Species (ROS)

Oxidative stress was determined by measuring the level of hydrogen peroxide (H_2_O_2_) generated in the cells, as assessed by 5-(6)-carboxy-2′,7′-dichlorodihydrofluorescein diacetate (carboxy-H_2_DCFDA) staining. The nonpolar, nonionic H_2_-DCFDA is cell permeable and is hydrolyzed to nonfluorescent H_2_-DCF by intracellular esterases. In the presence of peroxide, H_2_-DCF is rapidly oxidized to highly fluorescent DCF. In brief, at the end of test compound treatment, cells (2 × 10^5^) were washed with PBS and incubated with 5 μM H_2_DCFDA in DMSO for 30 min. The cells were then collected by trypsinization and centrifugation, washed with PBS, centrifuged at 200× *g* for 5 min and analyzed immediately using a Beckman Coulter FC500 flow cytometer.

### 4.10. Western Blot Analysis

Cell extracts were prepared in lysis buffer containing 20 mM Tris-HCl pH 7.4, 100 mM NaCl, 5 mM EDTA, 2 mM PMSF, 10 ng/mL leupeptin, and 10 μg/mL aprotinin. Equal amounts of extracted proteins (40 µg) were resolved by SDS-PAGE and transferred to nitrocellulose membranes (Schleicher & Schuell, Keene, NH, USA). The membranes were blocked with nonfat milk solution for 1 h, and then washed and probed with the appropriate primary antibody. The membranes were rinsed with Tris-buffered saline containing 0.1% Tween 20, incubated with horseradish peroxidase-conjugated secondary antibody for 1 h, rinsed again, and developed using enhanced chemiluminescence (ECL) reagents (Amersham Biosciences, Piscataway, NJ, USA).

### 4.11. Statistics

All data are expressed as the means ± SEs of three or more independent experiments. Between-group comparisons were performed using one-way analysis of variance (ANOVA) followed by an appropriate post-hoc test. A value of *p* < 0.05 was considered to be statistically significant.

## 5. Conclusions

In the present study, we focused on the apoptotic pathways that are induced by selected novel anthraquinone derivatives in human hepatocellular carcinoma cells that differ in their p53 status. In p53-wild-type HepG2 cells, binding of the drug to tNOX was found to interfere with tNOX-NAD^+^-SIRT1 activation, leading to p53 acetylation/activation and apoptosis. In p53-deficient Hep3B cells, on the other hand, inhibition of the tNOX-NAD^+^-SIRT1 axis increased c-Myc acetylation, which up-regulated TRAIL and DR5 expression to induce apoptosis. Based on these findings, we proposed that targeting of tNOX may be a potential strategy for cancer therapy in both p53-dependent and -inactivated systems and the tested compounds can be explored further as therapeutics against cancer.

## Figures and Tables

**Figure 1 cancers-11-00420-f001:**
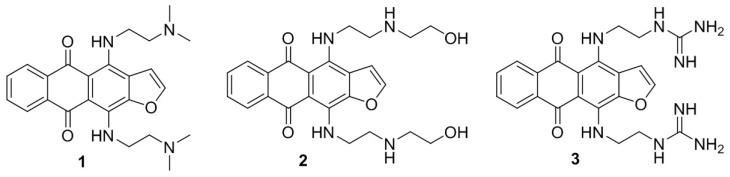
Structure of 4,11-diaminoanthra[2,3-*b*]furan-5,10-dione derivatives **1**–**3**.

**Figure 2 cancers-11-00420-f002:**
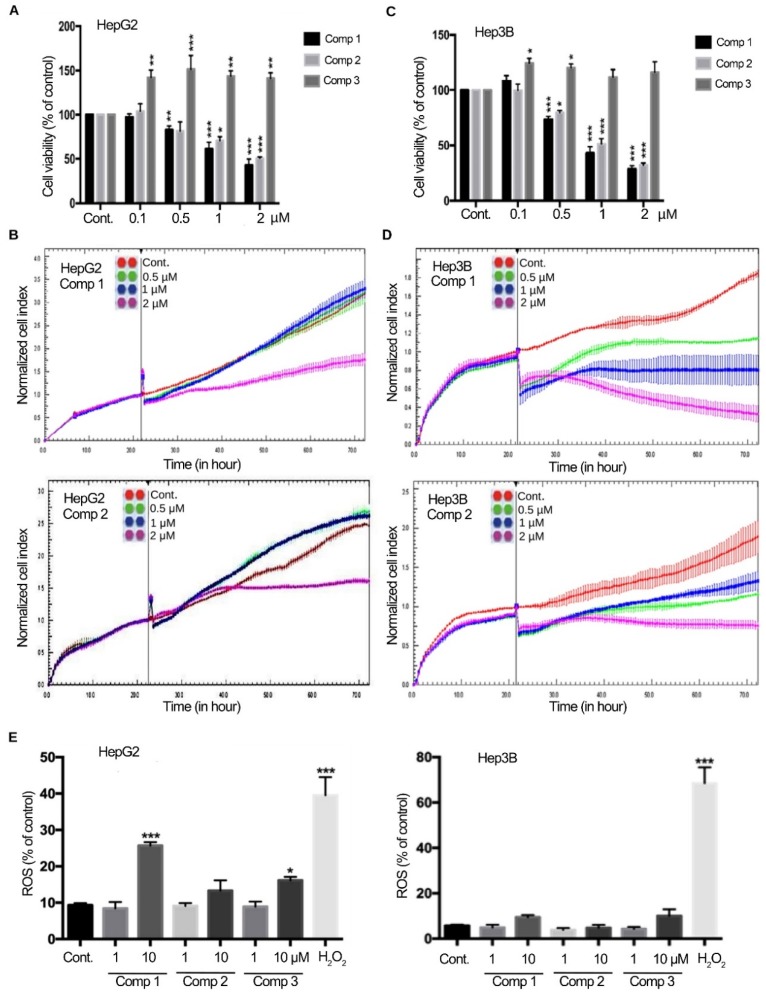
Selected derivatives suppress human hepatocellular carcinoma (HCC) cell viability and growth. (**A**,**C**) Cells were exposed to different concentrations of compounds for 24 h and cell viability was measured using MTS assays (HepG2 in **A** and Hep3B in **C**). Values (means ± SDs) were obtained from at least three independent experiments. There was a significant decrease in cell viability in cells treated with compounds compared with controls (* *p* < 0.05, ** *p* < 0.01, *** *p* < 0.001). (**B**,**D**) Cells were treated with compounds **1** and **2** and cell growth was dynamically monitored using impedance technology (HepG2 in **B** and Hep3B in **D**). Normalized cell index values measured over 70 h are shown. (**E**) The percent change in intracellular reactive oxygen species (ROS) generation was measured after cells were treated for 1 h with selected derivatives.

**Figure 3 cancers-11-00420-f003:**
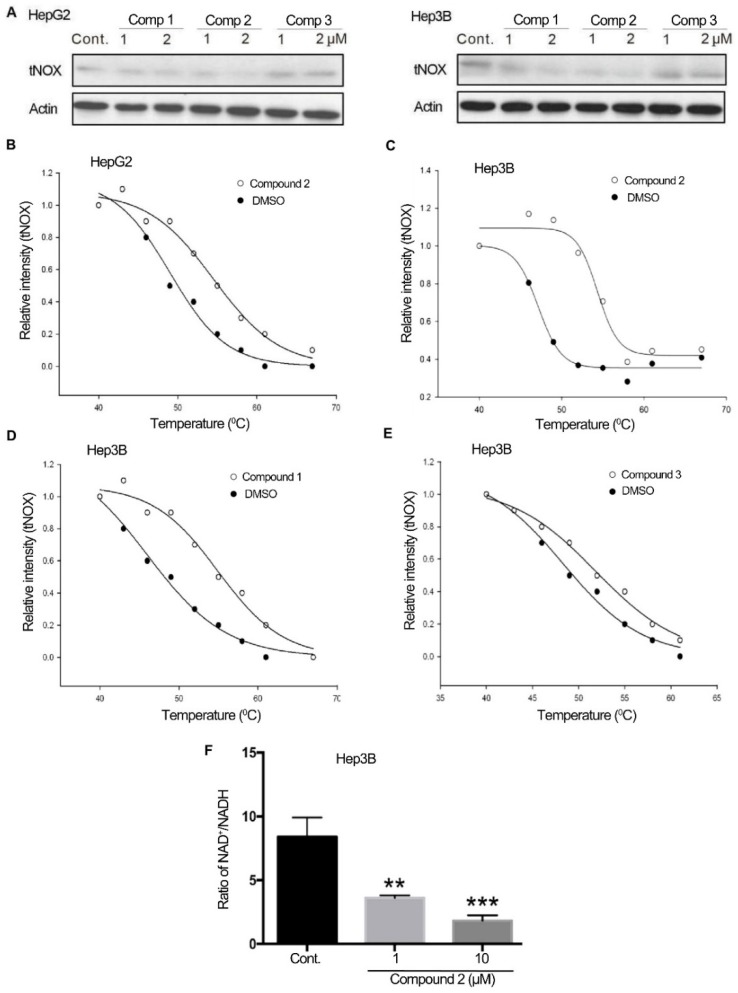
Effects of selected derivatives on tNOX expression and CETSA-based determination of binding between those compounds and tNOX. (**A**) Cells were treated with different concentrations of compounds or vehicle for 24 h, and cell lysates were separated by SDS-PAGE and analyzed by Western blotting. (**B**–**E**) CETSA curves of tNOX in p53-wild-type HepG2 cells (**B**) and p53-deficient Hep3B cells (**C**,**D**,**E**) were determined in the absence and presence of different compounds, and cell lysates were separated by SDS-PAGE and analyzed by western blotting. β-Actin was used as an internal control. Representative images are shown. The band intensities of tNOX were normalized with respect to the intensity at 40 °C. The denaturation midpoints were determined using a standard process. (**F**) Derivative **2** dose-dependently reduces the NAD^+^/NADH ratio in Hep3B cells. Cells were treated with vehicle or compound **2**, the NAD^+^ and NADH levels in cell extracts were quantified based on the optical density at 450 nm, and the NAD^+^/NADH ratio was calculated. Values (mean ± SEs) were obtained from at least three independent experiments.

**Figure 4 cancers-11-00420-f004:**
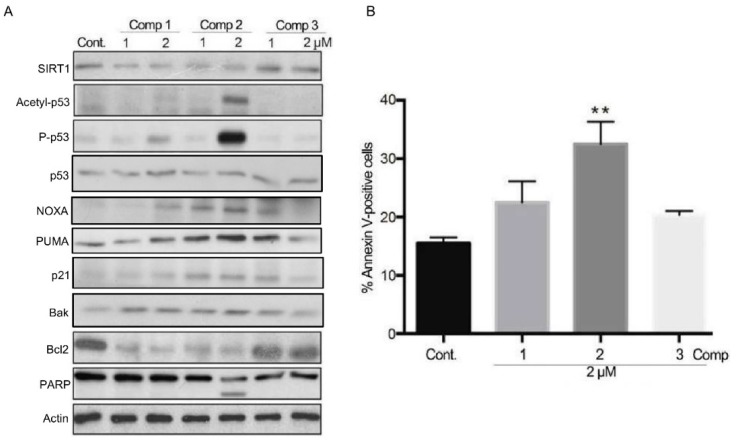
The effects of the selected derivatives on protein expression and apoptosis in HepG2 cells. (**A**) HepG2 cells were treated with different concentrations of compounds or vehicle for 24 h, and cell lysates were separated by SDS-PAGE and analyzed by western blotting. β-Actin was used as an internal control. Representative images are shown. (**B**) Cells were treated with different derivatives for 24 h, and the percentage of apoptotic cells was determined by flow cytometry. The presented values (mean ± SEs) represent as the percentage of cells in early and late apoptotic populations at least three independent experiments (** *p* < 0.01 for treated cells vs. controls). Representative images are shown.

**Figure 5 cancers-11-00420-f005:**
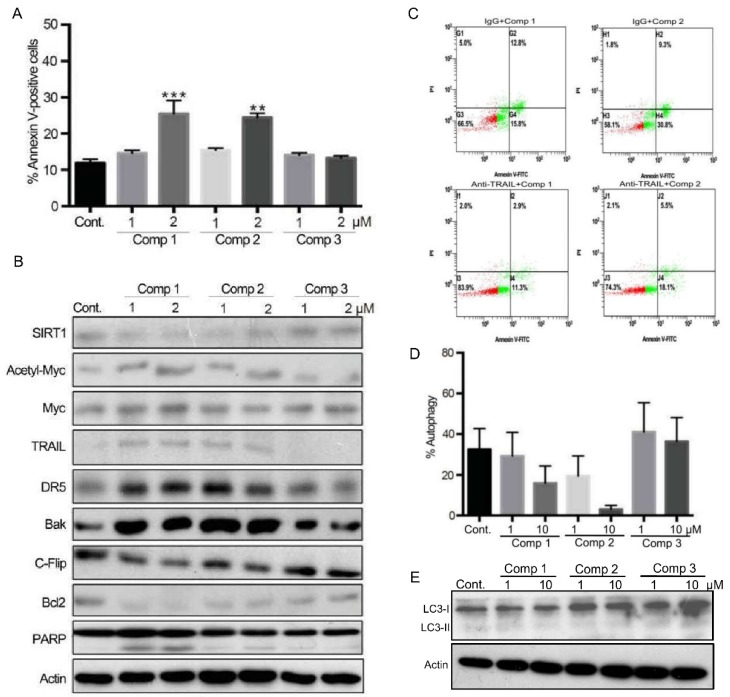
The effects of derivatives on protein expression and cell death in Hep3B cells. (**A**) Cells were treated with different derivatives for 24 h, and the percentage of apoptotic cells was determined by flow cytometry. H_2_O_2_ treatment was used as a positive control. The presented values (mean ± SEs) represent as the percentage of cells in early and late apoptotic populations at least three independent experiments (** *p* < 0.01, *** *p* < 0.001 for treated cells vs. controls). Representative images are shown. (**B**) Hep3B cells were treated with different concentrations of compounds or vehicle for 24 h, and cell lysates were separated by SDS-PAGE and analyzed by western blotting. β-Actin was used as an internal control. Representative images are shown. (**C**) Cells were pre-treated with IgG or TRAIL-neutralizing antibody at final concentration of 1 μL/mL for one h before exposed to different derivatives for 24 h. The percentage of apoptotic cells was determined by flow cytometry as described in **A**. (**D**) Cells were treated with different compounds or vehicle for 6 h. Autophagy was then determined by AO staining using flow cytometry, and the results are expressed as a percentage relative to the control group. Values (mean ± SEs) were obtained from three independent experiments. (**E**) Hep3B cells were treated with different concentrations of compounds or vehicle for 24 h, and cell lysates were separated by SDS-PAGE and analyzed by western blotting. β-Actin was used as an internal control. Representative images are shown.

**Figure 6 cancers-11-00420-f006:**
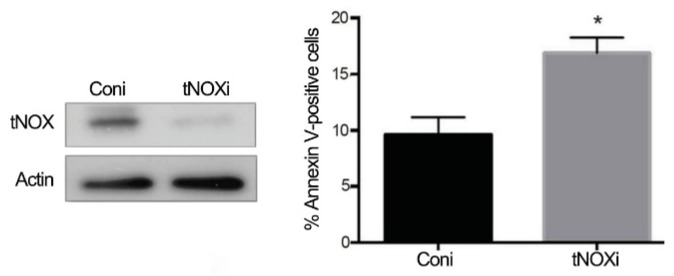
tNOX depletion enhances spontaneous apoptosis in Hep3B cells. tNOX was knocked down by RNA interference in p53-deficient Hep3B cells for 24 h, and the percentage of apoptotic cells was determined by flow cytometry. The presented values (mean ± SEs) represent at least three independent experiments (* *p* < 0.05 for tNOX-knockdown cells vs. controls). Representative images are shown.

**Figure 7 cancers-11-00420-f007:**
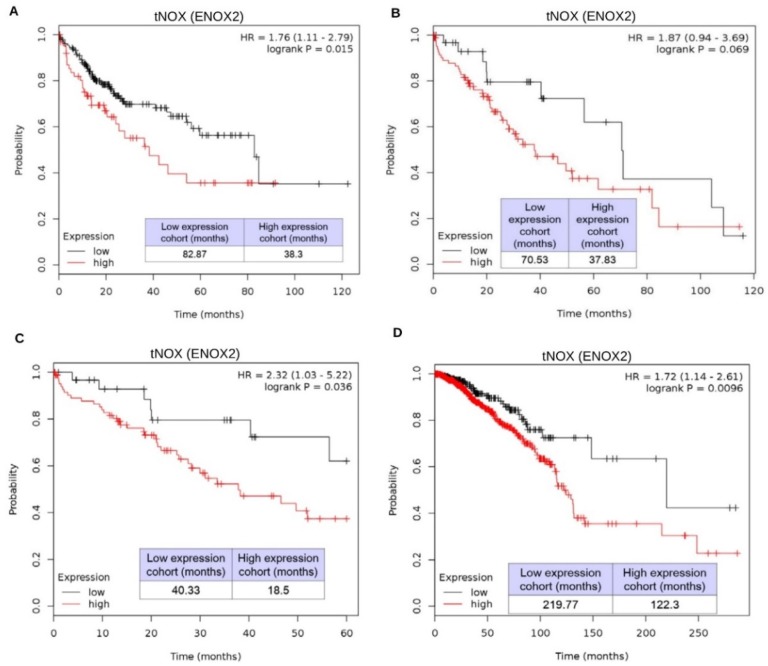
tNOX depletion enhances spontaneous apoptosis in Hep3B cells. tNOX was knocked down by RNA interference in p53-deficient Hep3B cells for 24 h, and the percentage of apoptotic cells High tNOX (ENOX2) expression is associated with worse survival outcomes in liver HCC patients. (**A**–**C**) Kaplan-Meier plots of the association between ENOX2 expression and overall survival in male (**A**) and female (**B**,**C**) liver cancer patients. (**D**) Kaplan-Meier plots of the association between ENOX2 expression and overall survival in female breast cancer patients. Data were obtained using Kaplan-Meier Plotter.

**Figure 8 cancers-11-00420-f008:**
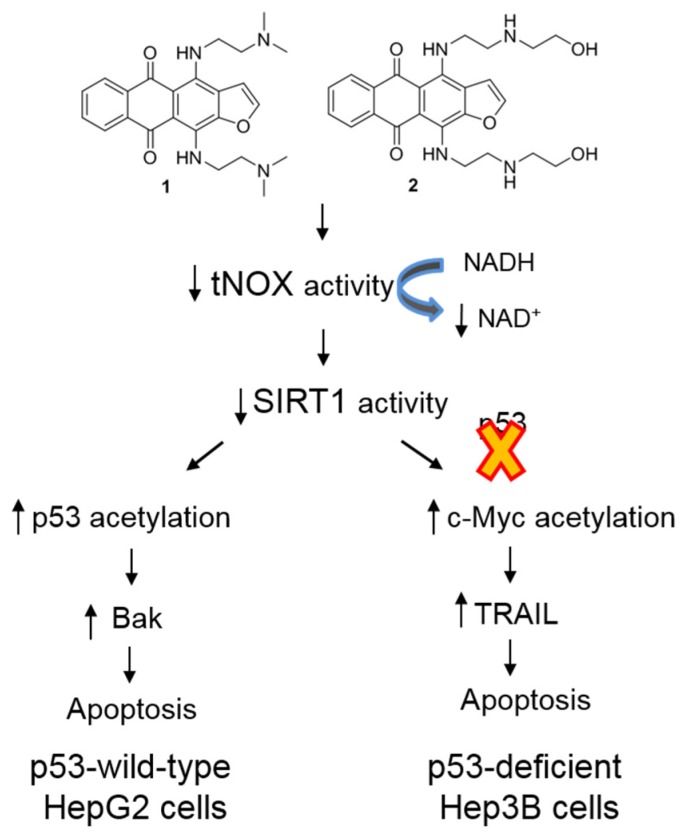
Schematic diagram of the pathways that lead to the different pathways of p53-wild-type HepG2 and p53-deficient Hep3B cells by 4,11-diaminoanthra[2,3-*b*]furan-5,10-diones **1** and **2**.

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
