# Peer review of "Engagement with tNOX (ENOX2) to Inhibit SIRT1 and Activate p53-Dependent and -Independent Apoptotic Pathways by Novel 4,11-Diaminoanthra[2,3-*b*]furan-5,10-diones in Hepatocellular Carcinoma Cells"

_cancers, 2019, doi:10.3390/cancers11030420_

Reviewer 1 Report

                In this manuscript, the authors determined the mechanism of apoptosis in hepatocellular carcinoma (HCC) cell lines induced by new anticancer compounds synthesized by them. In p53 wild type HCC cells, compound 1 and 2 decreased SIRT1, resulting in enhanced acetylation and activation of p53. In contrast, in p53 mutant HCC cells, compound 1 and 2 enhanced acetylation of c-Myc and expression of TRAIL. Therefore, the authors suggest that these compounds induce apoptosis mediated by p53 in p53-wild HCC and by TRAIL in p53-mutant HCC cells. Although the authors have reported that related compounds showed cytotoxicity in various cancer cells as described in the text, but this is the first time that these compounds induced apoptosis in HCC cells using two different pathways. Therefore, I think that this manuscript is worth publishing in Cancers.

My specific comments are as follows.

1. In Fig. 4A, the authors demonstrated that the SIRT1 expression was diminished by compounds 1 and 2, and that p53 acetylation and phosphorylation were enhanced in HepG2 cells. At the same time, the expression of apoptosis-inducer Bak and Bax was activated by these compounds. However, the expression of Bak, Bax, and also p53 and the degree of p53 acetylation/phosphorylation do not seem to correlate. To investigate whether the increase in p53 acetylation and phosphorylation is related to the induction of apoptosis, the authors should show the transcriptional activity of p53 by the expression of p53-inducible genes, such as p21, PUMA and Noxa.

2. In Fig. 4B and 5B, t he authors should describe which fraction of FITC/PI staining was identified as% Annexin V-positive cells.

3. Since it is not adequate to discuss the degree of autophagy only with AO staining, the authors should detect the ratio of LC3-I and LC3-II by immuno blot.

4. Since it can be easily identified using neutralizing antibody that induction of TRAIL is involved in apoptosis, the authors should perform this experiment.

Author Response

In this revision, we have repeated some of the experiments and generated new results in response to Reviewer #1’s suggestions. All the modifications made in the revised manuscript are highlighted in blue.  

Reviewer #1

1.     In Fig. 4A, the authors demonstrated that the SIRT1 expression was diminished by compounds 1 and 2, and that p53 acetylation and phosphorylation were enhanced in HepG2 cells. At the same time, the expression of apoptosis-inducer Bak and Bax was activated by these compounds. However, the expression of Bak, Bax, and also p53 and the degree of p53 acetylation/phosphorylation do not seem to correlate. To investigate whether the increase in p53 acetylation and phosphorylation is related to the induction of apoptosis, the authors should show the transcriptional activity of p53 by the expression of p53-inducible genes, such as p21, PUMA and Noxa.

      Reply: We thank Reviewer 1 for the suggestion. In revised Figure 4, we added p21,        PUMA, as well as Noxa, and Bax is removed from Figure 4.

2.     In Fig. 4B and 5B, the authors should describe which fraction of FITC/PI staining was identified as% Annexin V-positive

Reply: In both apoptosis determinations, we assigned the % Annexin V-positive as both early apoptotic (FITC-positive) and late apoptotic (FITC/PI double-positive). We have added the description accordingly in the revised version.

3.     Since it is not adequate to discuss the degree of autophagy only with AO staining, the authors should detect the ratio of LC3-I and LC3-II by immune blot.

Reply: Western blot of LC3-I and LC3-II are provided in revised Figure 5. The lack of cleaved LC3-II supports the notion that compound 1 and 2 do not induce autophagy. Due to those new results, we have reorganized the content of Figure 5 and manuscript is revised accordingly.

4.     Since it can be easily identified using neutralizing antibody that induction of TRAIL is involved in apoptosis, the authors should perform this

Reply: We appreciate this suggestion from reviewer 1. Anti-TRAIL antibody was used to neutralize the apoptotic function of TRAIL ligand and our results show that the neutralizing antibody is effective in the attenuation of apoptosis induced by compound 1 and 2. Those new results are now in revised Figure 5C.

Reviewer 2 Report

Hepatocellular carcinoma is the most frequently occurring primary malignancy of the liver and one of the top three causes of cancer-related death. Treatment for hepatocellular carcinoma includes surgical resection, liver transplantation, percutaneous radiofrequency ablation, with or without chemotherapy. The human gene tNOX located on Xq25-26 is universally expressed in an array of cancer cells. TNOX is associated with cell proliferation and migration and the suppression of tNOX induces apoptosis and attenuates cancer cell growth. There has been a recent synthesis of anti-cancer compounds based on furan-fused anthraquinone derivatives with different side chains, which have shown high antiproliferative activities against various cancer cell lines.  The aim of this study was to investigate three selected 4,11-diaminoanthra [2,3-b] furan-5,10-dione derivatives for their toxicity and underlying mechanisms in wild-type hepatocellular carcinoma cells.  Two liver cancer cell lines; HepG2 (p53 wild type) and Hep 3B (p-53 deficient) were used in this study, as well as, an online database containing 370 patients with hepatocellular carcinoma. The researchers used a variety of investigations for this study including cell viability assay, apoptosis determination, intracellular NAD+/NADH ratio, cellular thermal shift assay (CETSA), measurement of reactive oxygen species, western blotting and statistical analyses.  This study is a follow up to a previous study done by the researchers in which they demonstrated that 4,11-diaminoanthra [2,3-b] furan-5,10-diones induced apoptosis.

Novelty /Originality

This article is sufficiently novel and interesting to warrant publication. No previous studies were found for engagement of tNOX to inhibit SIRT1 and activate p53 dependent and independent apoptotic pathways by novel 4,11-Diaminoanthra (2,3-b) furan-5,10-diones in hepatocellular carcinoma cells This article can contribute to the advancement of science and the delivery of healthcare as it has the potential to improve the management of hepatocellular carcinoma.

Presentation 

This article was clearly laid out with all the key elements present. The title clearly described the content of the article, while the abstract provided a good summary of the content of the manuscript. In the introduction, the authors clearly stated their objectives and the aim of their investigation. The methodology used and results obtained were clearly described by the authors. However, the authors chose to have the methodology section after their discussion and just before the conclusion. The methodology would be better if it appeared after the introduction and before the results. The study design was suitable for the aim of the study with adequate statistical analysis conducted on the results obtained. Appropriate graphs and pictures which were both clear and informative were included in the manuscript. In the discussion, the author summarized his findings with these findings being relevant to previous studies. The results obtained supported the claims of the researcher with the speculations and extrapolations being reasonable. The article was well written and the language used was scientific.

Importance

The study demonstrated 4,11-Diaminoanthra (2,3-b) furan-5,10-diones bind to tNOX and inhibited tNOX-catalyzed NAD generation. It also inhibited SIRT1 deacetylase activity while enhancing p53/c-myc acetylation leading to apoptosis. The expression of tNOX is essential for Hep 3B cell survival and correlates with tumor progression and poor prognosis. The findings of this study have the potential to improve the management of hepatocellular carcinoma and tNOX may be a potential therapeutic target against cancer.

References

The references used in this manuscript were sufficient, appropriate and recent.

Scientific Merit

Hepatocellular carcinoma is the most frequently occurring primary malignancy of the liver and one of the top causes of cancer-related death worldwide. The study demonstrated that compounds 1 and 2 of 4,11-diaminoanthra [2,3-b] furan-5,10-dione engaged with tNOX and inhibited SIRT1 to induce apoptosis against both p53 wild type and p53 deficient hepatocellular carcinoma cell lines. Binding of the drug to tNOX was found to interfere with tNOX-NAD+-SIRT1 activation, which led to p53 acetylation/activation and apoptosis in the p53-wild type Hep G2 cells. While inhibition of the tNOX-NAD+-SIRT1 axis increased c-Myc acetylation which upregulated TRAIL and DR5 expression, thereby inducing apoptosis in p53 deficient Hep 3B cells. This information and further research can potentially provide an alternative treatment option for patients with hepatocellular carcinoma. This study was warranted and the findings of this study can provide a better understanding of the mechanism of apoptosis via the tNOX-SIRT1 axis. Recommendations such as the need for further research were addressed by the researchers.

In the discussion, the impact of the findings at the subcellular level should be discussed, even the researchers did not present ultrastructural data.

Ethical Issues

There was neither plagiarism nor fraud in this manuscript. 

Author Response

Reviewer #2
1)     In the discussion, the impact of the findings at the subcellular level should be discussed, even the researchers did not present ultrastructural data.

     Reply: We appreciate all valuable comments from Reviewer 2. In response to this comment, we have added a statement of the impact of our findings on page 13 as in “These novel anthraquinone-derived compounds inhibit tNOX-NAD+-SIRT1 axis to induce apoptosis, highlighting the value of tNOX as a possible therapeutic target and implying future clinical use of these compounds in cancer treatment without considering p53 expression.”

Round  2

Reviewer 1 Report

I think that this revised manuscript is now acceptable for publication in Cancers.